# High-Intensity Training Reduces CVD Risk Factors among Rotating Shift Workers: An Eight-Week Intervention in Industry

**DOI:** 10.3390/ijerph17113943

**Published:** 2020-06-02

**Authors:** Asgeir Mamen, Reidun Øvstebø, Per Anton Sirnes, Pia Nielsen, Marit Skogstad

**Affiliations:** 1School of Health Sciences, Kristiania University College, Box 1190, Sentrum, 0107 Oslo, Norway; 2The Blood Cell Research Group, Department of Medical Biochemistry, Oslo University Hospital, 0450 Ullevaal, Norway; reidun.ovstebo@medisin.uio.no; 3Østlandske Hjertesenter, 1523 Moss, Norway; pas@cardio.no; 4Ringvoll BHT, 1523 Moss, Norway; pia.nielsen@ringvollbht.no (P.N.); marit.skogstad@ringvollbht.no (M.S.)

**Keywords:** shift work, cardiovascular, occupational health, physical activity

## Abstract

Rotating shift work is associated with risk factors for cardiovascular disease (CVD). We have studied the effect of 17 min high-intensity training three times a week over eight weeks on CVD risk factors among shift workers. Sixty-five shift workers from two plants were recruited. They were all deemed healthy at the initial health screening and in 100% work. From plant A, 42 workers, and plant B, 23 workers participated. After the intervention, 56 workers were retested. The intervention group consisted of 19 participants from plant A who had participated in at least 10 sessions. Twenty workers from plant B and 17 workers from plant A that not had taken part in the training were included in the control group. All workers reported physical activity (PA) by questionnaires before and after the training intervention. We measured blood pressure, heart rate, lipids, glycated hemoglobin (HbA1c), and C-reactive protein (CRP) and arterial stiffness. Maximal oxygen uptake ($\dot{V}$O_2max_) was assessed by bicycle ergometry. The intervention group favorably differed significantly from the control group in improvement of systolic and diastolic blood pressure and glycated hemoglobin (HbA1c). Short training sessions with 4 min of high-intensity PA, three times a week, for eight weeks among rotating shift workers reduced some CVD risk factors. PA interventions in occupational settings may thus decrease coronary heart disease and stroke incidences in this vulnerable group of workers.

## 1. Introduction

Economic globalization affects the labor market and work organizations. This could result in more use of long working hours and night shifts [1]. In Europe, 21 % of the workforce is engaged in shift work and 19% work at night [2]. An increased risk for cardiovascular disease (CVD) among shift workers has been known for many decades. Recent systematic reviews and meta-analysis encompassing more than 2 million individuals reveal an association between shift work and CVD [3,4], even demonstrating CVD events increasing by 7.1% for every five years of exposure after the first five years as a shift worker [4].

CVD risk factors in shift workers could be due to disturbances of the endocrine system including unfavorable catecholamine, ghrelin (“hunger hormone”), and adipokine excretion, dyslipidemia, metabolic syndrome, insulin resistance and melatonin reduction, and ultimately diabetes [5,6], but also an increased systemic inflammation [7,8,9]—even due to circadian misalignment per se [10]—and changes to the immune system [11].

In a study running for three years, we initially examined early manifestation of CVD risk factors in shift workers in two industrial plants in Norway [12]. We studied blood parameters including lipids (cholesterol, low-density lipoprotein (LDL), and high-density lipoprotein (HDL)), glycated hemoglobin (HbA1c), C-reactive protein (CRP), brachial blood pressure, resting heart rate and central blood pressure, augmentation pressure and index, pulse wave velocity, and ultrasound measurement of the carotid arteries. In addition, we assessed maximal oxygen uptake ($\dot{V}$O_2max_) [12]. 

The shift workers reported less physical activity (PA) of a high degree of intensity compared to the day workers [12], but the $\dot{V}$O_2max_ was within expected values for a corresponding population of Norwegians [13]. In this cross-sectional study of workers in industry, the number of years of shift work was associated with early manifestations of CVD, with an increased intima media thickness (IMT) in the carotid artery and increasing CRP. This could imply an increased risk for coronary heart disease and stroke among these workers. We will follow the present cohort for three years. 

As for CVD risk factors, we found increased carotid intima media thickness (cIMT) and CRP associated with number of years as a shift worker [12]. In order to reduce the cancer risk of shift work, a concern for lack of knowledge, practical advice, and evaluated interventions for workers has been voiced [14]. The same applies when it comes to reducing risk for CVD. Longitudinal studies on shift work in industry and PA are scarce and generally cross-sectional studies are not conclusive when it comes to assessing differences between shift workers in industry and day workers on engagement in PA during leisure time [15]. Furthermore, there is a lack of PA intervention studies among shift workers. One study reports increased $\dot{V}$O_2max_ along with decreased heart rate after a four-month intervention program among female shift workers working day, evening, and night shifts [16]. Otherwise, we have not detected interventions addressing CVD risk factors in this group of workers.

High-intensity PA (e.g., running once a week or 50 min/week) has been shown to reduce the risk of all-cause and CVD mortality [17]. Generally, high-intensity training affects early manifestation of CVD such as lowering blood pressure and affecting blood lipids and blood glucose favorably, but also decreasing systemic inflammation [18]. 

Aim of the study: Given our results of CVD risk factors among shift workers [12] and a worry that shift workers tend to exercise less than other workers [6], the Occupational Health Service wanted to study if high-intensity PA could modify the risk of early manifestations of CVD in this group of industrial workers. 

## 2. Material and Methods 

We collected background data and medical history by questionnaire. We have described the present cohort in a recent paper [12], but in short, we recruited 94 participants from two insulation-producing plants (A + B) in Norway. From plant A, 42 out of 51 shift workers agreed to participate (82%), and from plant B, the corresponding figures were 23 of 55 shift workers (42%). Here, 29 dayworkers (75 eligible) were also included in the cohort. Thirteen of the 94 workers were female (14%) (Table 1). See also Table 2 for baseline description of the two groups.

The shift workers worked rotating shifts with day, evening, and night shifts even lasting for 12 h [12]. An intervention group consisted of volunteers from plant A, and the workers from plant B made up the control group. Seventeen of the workers in plant A preferred to do PA by themselves and were thus analyzed among the control group; see Figure 1.

The examinations took place in August 2018 and then after an eight-week PA initiative during autumn 2018 and spring 2019. The Regional Ethics Committee in Oslo approved of the study (2018/1258). We informed the participants about the study and they gave their written consent to participate (ISRCTN42416837).

While the participants were sitting, we measured blood pressure and resting heart rate (RHR) to look for signs of cardiovascular disease. The measurements took place after five minutes of rest, on the left arm three times in intervals of one minute, and the mean values of three measurements of the systolic (SBP) and diastolic pressure (DBP) were used in the statistical analysis in both occasions. Using the mean is the current practice at our laboratory. We used a BpTRU Vital Signs Monitor™ BPM-100 (Bp TRU Medical Devices, Coquitlam, BC, Canada).

We assessed central blood pressure (CBP), which is the pressure in the ascending aorta, augmentation pressure (AP), augmentation index (AIx), pulse pressure (PP), and pulse wave velocity (PWV) with a SphygmoCor XCEL^®^ (AtCor Medical Pty Ltd., Sydney, Australia). The participants were in supine position during this examination and we used HR from this examination as RHR when setting up the HR monitor. We performed the measurements according to the manufacturer’s recommendations (www.atcormedical.com). 

Blood for serum investigations of glycated hemoglobin (HbA1c) and lipids (cholesterol (CHOL), low-density lipoprotein (LDL), high-density lipoprotein (HDL)) was drawn from the participants using ethylene diamine tetra acetic acid (EDTA) tubes. We used whole blood gel tubes for C-reactive protein (CRP). The tubes were centrifuged at 30 × 1000 RPM for 10 min within 60 min of the blood being drawn from a vein. Samples were transported to the Department of Medical Biochemistry, Oslo University Hospital and analyzed within 48 h. HbA1c (EDTA blood) was analyzed with a Tosoh G7 HPLC analyzer (Tosoh Bioscience, Inc. San Francisco, CA, USA) using the “high-performance liquid chromatography” separation principle. The analytical variation was 1.7%. CHOL, LDL, and HDL in serum were analyzed by enzymatic colorimetric method in a Cobas 8000 Modular Analyzer (Hoffmann-La Roche, Basel, Switzerland). Analytical variation coefficients were 3.0%, 4.0%, and 3.5%, respectively. CRP was assessed in serum by particle enhanced immunoturbidimetric method on the Cobas 8000. The analytical variation was 8.0%. 

We tested the participants with a graded exercise test on a cycle ergometer (Monark 874E, Monark Exercise AB, Vansbro, Sweden). The starting load was 70 W with a cadence of 70 RPM. The subjects were instructed to maintain a cadence of between 68 and 72 RPM during the test. Every minute, the resistance was increased by 28 W (0.4 Kg), until voluntary exhaustion or if the cadence dropped below 60 RPM despite verbal encouragement to increase cadence. The subject wore a heart rate belt and transmitter from Polar (Polar OY, Kempele, Finland). The signal from this transmitter was presented on the bike’s screen and recorded every minute during the test. A K5 metabolism analyzer (Cosmed Srl, Rome, Italy) was attached to the test subject through a Hans Rudolph 7400 Vmask oro-nasal face mask (Hans Rudolph Inc, Shawnee, KS, USA), which was fitted to the subject and then checked for leakage before the test started. Oxygen uptake was measured continuously with the K5 using the unit’s mixing chamber and 10 s measurement intervals. $\dot{V}$O_2max_ was defined as the median of the three highest successive 10 s measurements at the end of the test. Criteria for a valid test included a respiratory exchange ratio of >1.05 or heart rate >95% of age predicted maximal heart rate, together with the test leader’s evaluation of fatigue. Time to exhaustion and end load were also recorded. The highest heart rate recorded, plus five bpm, was considered maximal heart rate (HR_max_) [19] and used when setting the maximal heart rate in the heart rate device. This value was so increased by 5–10 bpm for running HR_max_. 

We assessed physical activity (PA) by the question: “How often do you normally exercise?”, with the response alternatives of “never or less than once per week”, “once per week”, “two to three times per week”, or “almost every day” [20]. Furthermore, we asked participants to report, in minutes per week, the total amount they were engaged in PA with high intensity (e.g., running, Spinning^®^). Physical activity was also measured by a Mio Slice wrist-worn heart rate monitor (Mio Global Inc., Vancouver, Canada) that calculated PA from heart rate and personal information and presented it as Personalized Activity Intelligence (PAI) [21]. All participants were asked to acquire 100 PAI per week.

An eight-week PA intervention with supervised PA by an experienced occupational physiotherapist was organized by the Occupational Health Service. The company had provided a gym at the plant with treadmills, bicycles, and row machines. The workers met before or after a work shift for PA. One responsible person in each shift recorded the presence of the participants in a compliance form. We adapted the PA program by Tjønna et al. [22] lasting for 17 min in total. The main phase was a four-minute work session at near maximal effort (~90% of HR max). This was performed three times a week. Except for ultrasound measurement of the carotid artery, all measurements at the eight-week follow-up were similar to that of baseline registrations [12].

**Statistical analysis.** Descriptive data are presented as mean with (SD) unless otherwise noted. The differences in change between intervention and control group, and between baseline and post-intervention results were assessed with Gosset (Student) independent/paired t-tests and with Effect Size (Cohen’s d with Hedge’s g correction). Analyses were carried out using SPSS v. 26 (IBM SPSS, Armonk, NY, USA) and Stata IC 16 (Stata Corp LLC, College Station, TX, USA). Cohen’s d effect size with Hedge’s g correction (ES) analyses were done with an Excel spreadsheet (http://www.cemcentre.org/renderpage.asp?linkID=30325017). Bar graphs show mean and SD. In box plots, the whiskers are the ±95 percentiles and the horizontal line the median.

**Ethical considerations.** The Regional Ethics Committee in Oslo approved of the study (2018/1258). We informed the participants about the study and they gave their written consent to participate (ISRCTN42416837).

## 3. Results 

In Table 3, the development from baseline (BL) to post-intervention (PO) is shown for the two groups. The control and intervention groups both develop favorably from BL to PO on SBP, DBP, aorta systolic blood pressure (ASBP), PP, AP, CHOL, HbA1c, and Vigorous Physical Activity (VPA). In addition, the intervention group had a positive development on body mass (BM), aorta diastolic blood pressure (ADBP), CRP, and LDL. The control group had a positive development of r$\dot{V}$O_2max_ and a$\dot{V}$O_2max_ not present in the intervention group. PWV developed negatively in both groups, and the control group also had a negative development of BM, ADBP, CRP, and LDL. The development in BM and HbA1c was statistically significant for the control group (*p* < 0.05 and *p* < 0.01). In the intervention group, PO and BL values for ASBP, ADBP, and HbA1c were statistically significant (*p* < 0.05 for ASBP and ADBP and 0.01 for HbA1c). Concerning the size of change between control and intervention groups, they differed on BM and CHOL (*p* < 0.05). See Table 3.

Self-reported training frequency changed during the intervention. At baseline, a majority of the participants trained one or less than one session per week. After the intervention, all members (100%) of the intervention group reported training two to three times a week. The control group members reduced the number that trained one or less than one time per week after intervention and increased the numbers that reported training two or more times per week. Self-reported weekly VPA increased by 11.4 (83.2) min in the control group and 15.6 (116.1) min in the intervention group (Figure 2A,B). We had technical problems evaluating the PAI results, but the intervention group reached an average of 100 PAI per week during the intervention period, statistically significantly more than that of the control group (results not shown).

$\dot{V}$O_2max_ relative to body mass did not change much for either groups, but interestingly, the control group had a small increase in mean absolute $\dot{V}$O_2max_ as seen in Figure 3.

An ES analysis showed that for BM, ADBP, CHOL, and LDL, ESs were > 0.50, often described as clinically relevant, showing that the intervention group had in all a more favorable development.

Differences between the groups in blood pressure, cholesterol, arterial stiffness markers, and inflammation/diabetes markers are shown in Figure 4.

## 4. Discussion 

We found minor differences in important CVD risk factors between a group of rotating shift workers that had participated in an eight-week training intervention with 1x4 min high-intensity bouts, three times a week, and colleagues that had not followed this training, but kept their normal activity level. The intervention group favorably differed significantly from the control group on improvement of systolic and diastolic blood pressure and glycated hemoglobin (HbA1c). The groups’ post-intervention values were not statistically significantly different for any variable, but the *size* of the change was significantly different for BM and CHOL (*p <* 0.05), with the intervention group having the largest changes, in a positive direction.

A strength of the present paper is its prospective design in which the participants are their own control, eliminating the variation between subjects. This design also gives the opportunity to assess each participant’s health outcome after an intervention. The same researchers/technicians performed the tests at both occasions using the same devices and instructing the participants in the same manner. The participants were all rotating shift workers but still self-selection of healthy and highly motivated individuals into the study could not be ruled out. 

A limitation of the study could be misclassification of exercise level since the most motivated participants in the study could tend to report PA sessions more often than less compliant participants. Furthermore, stronger associations between PA and effects of health outcome are probable in laboratory settings with individuals not working as rotating shift workers. As the criterium for being admitted to the intervention group was only to have attended at least 10 sessions, the training effect in that group may have been low. Both the intervention group and the control group were asked to have an active life, and to accumulate 100 PAI per week, so the difference between the two groups in PA level was not maximized.

Self-reported training did increase for both groups (Figure 2). The control group reduced their number of less active participants (training one or less times per week) and increased the number in the more active groups. This is what they were asked to do and is not a John Henry effect. We wanted as many as possible to achieve 100 PAI per week through the intervention period as this may have a positive effect on cardiovascular health [23], but due to technical problems with the hardware, we were not able to use the results from this registration in further analysis, but a motivating factor of the use of these items could not be ruled out. For the intervention group, all participants (100%) reported training two to three times a week at intervention end, thereby reducing the number that had trained more than three times a week from 9% to 0%, but at the same time increasing the number of participants that trained less than three times per week substantially. This shows that the intervention group did follow the project’s training offer to a large extent, and that this training was regarded as sufficient and thus created no need for more training sessions, even if the training time per session was only 17 min in total. The participants were also asked to report the amount of VPA per week, and both groups reported increased activity, but the intervention group was at both times reporting a higher number of minutes engaged in VPA than the control group. Self-reported activity should be looked upon with caution, as it is easy to overlook training, especially low-intensity activity. As we asked about high-intensity activity, the answers may be more accurate [24,25,26]. 

Intervention studies have shown that those who engage in PA on a regular basis are more easily convinced to participate in PA initiatives compared to sedentary ones [27]. This may thus induce a bias in the sample that can make the effect larger than it really is. However, the participating rate was high in the intervention group as seen in Figure 1. We also know that sedentary workers were included in the present intervention as demonstrated in the case presentation, Table 4. This case had, with the exception of CRP, a favorable change in all parameters after eight weeks of training: resting heart rate, blood pressure, measures of arterial stiffness, blood sugar, and $\dot{V}$O_2max_. 

Among rotating shift workers, increased blood pressure [28], most pronounced among workers with mostly night work and frequent rotations [29], and arterial stiffness have been described [30,31]. In the present study, blood pressure decreased after an eight-week high-intensity training intervention focusing on cardiorespiratory performance. The reduction was small, but represents a change in the right direction. PA has a protective effect on stiffness of the arteries [32]. Both augmentation pressure and pulse wave velocity were associated with self-reported high-intensity PA at baseline and post-intervention in the present cohort. 

Low HDL is reported among shift workers [33,34,35] and associated with the metabolic syndrome. In addition, sleep deprivation downregulates cholesterol/lipid metabolism and transport [36]. In the present study, we found that regular weekly high-intensity PA for eight weeks yielded an elevation of HDL and a reduction of total cholesterol among those participating in guided PA (intervention group).

Disruption of the circadian rhythm increases the risk for diabetes and metabolic syndrome [28]. Two large cohort studies of female nurses have found that rotating night shift work as such is associated with diabetes II [37]. Any measure that could result in decreased risk for this disease, and thus counteract its cardiometabolic consequences, is essential. Therefore, regular high-intensity PA of short duration, leading to a reduced HbA1c as demonstrated in the present study, is of great importance as a CVD preventive strategy among rotating shift workers. 

Systemic inflammation expressed as elevated high-sensitivity CRP is known to increase vascular risk. Shift work could cause sleep deprivation which again affects the immune system, resulting in increased inflammation [36,38]. This phenomenon could explain that number of years as a shift worker is associated with increasing CRP levels initially in the present cohort [12]. Leisure time PA among workers with long working hours, however, resulted in reduced CRP after 15 months of follow-up [12]. The present eight-week PA intervention did not affect this parameter. Thus, the duration of this PA initiative might be too short to reduce inflammation among shift workers. 

Shift work is associated with increased carotid intima-media thickness (CIMT) and thus possibly early manifestation of atherosclerosis [12,39]. PA reduces CIMT progression among patients with diabetes II and coronary heart disease [40]. The present study suggests beneficial effects of PA educational programs on protection of early manifestation of CVD risk factors and atherosclerosis, as depicted in Table 5. The cohort will be subjected to three years of follow-up in which CIMT will be repeated at final follow-up.

## 5. Conclusions

In this study, we found that a PA intervention focusing on cardiorespiratory health among rotating shift workers in industry can modify early manifestations of CVD. This could imply that regular, short bouts of PA could counteract the increased risk for coronary heart disease and stroke among these workers. PA interventions in occupational settings may thus decrease coronary heart disease and stroke incidences in this vulnerable group of workers. The present cohort will be subjected to follow-up for three years.

## Figures and Tables

**Figure 1 ijerph-17-03943-f001:**
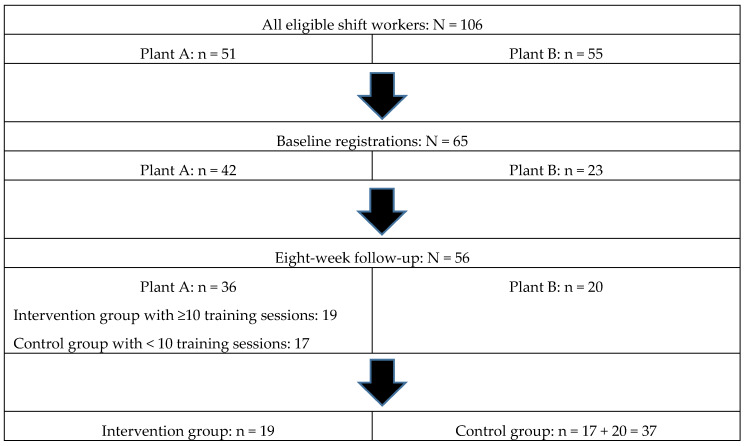
Selection process. 106 shift workers were available for the project, 51 from plant A and 55 from plant B. At project start, 42 shift workers from plant A accepted the invitation with three weekly trainings sessions, with 23 workers from plant B as “activity as usual” controls. At follow-up eight weeks later, 19 participants from the intervention group had participated in at least 10 training sessions and were included in the analyses as intervention group. The remaining 17, who had less than 10 training sessions logged, were put in the control group, together with the 20 workers from plant B, giving 37 in this group.

**Figure 2 ijerph-17-03943-f002:**
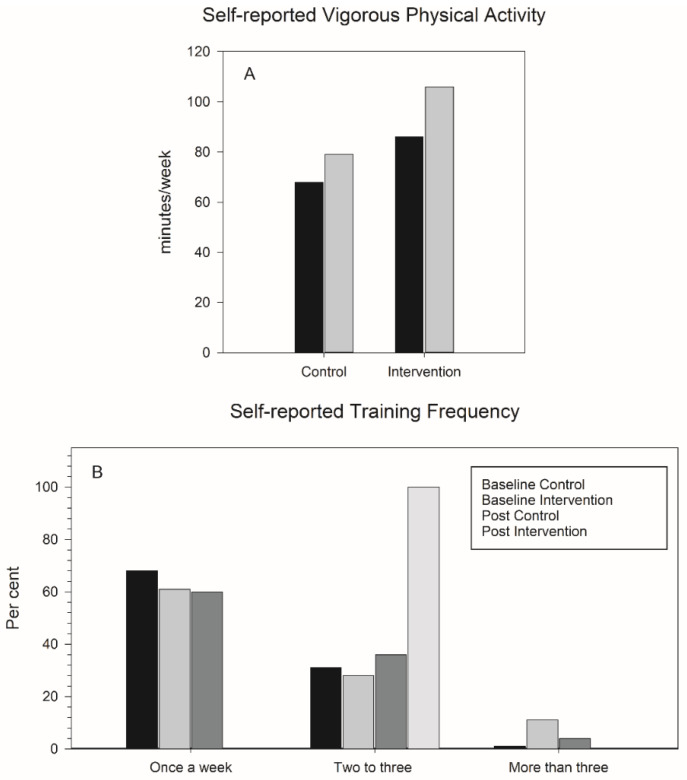
Self-reported change in VPA (**A**), self-reported training frequency (**B**).

**Figure 3 ijerph-17-03943-f003:**
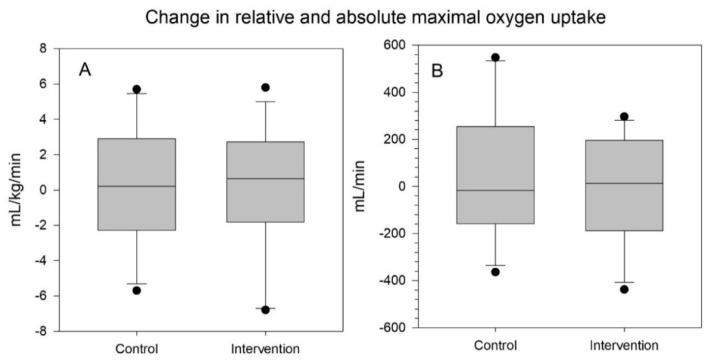
Change in maximal oxygen uptake, (**A**) relative to body mass, (**B**) absolute, in L/min.

**Figure 4 ijerph-17-03943-f004:**
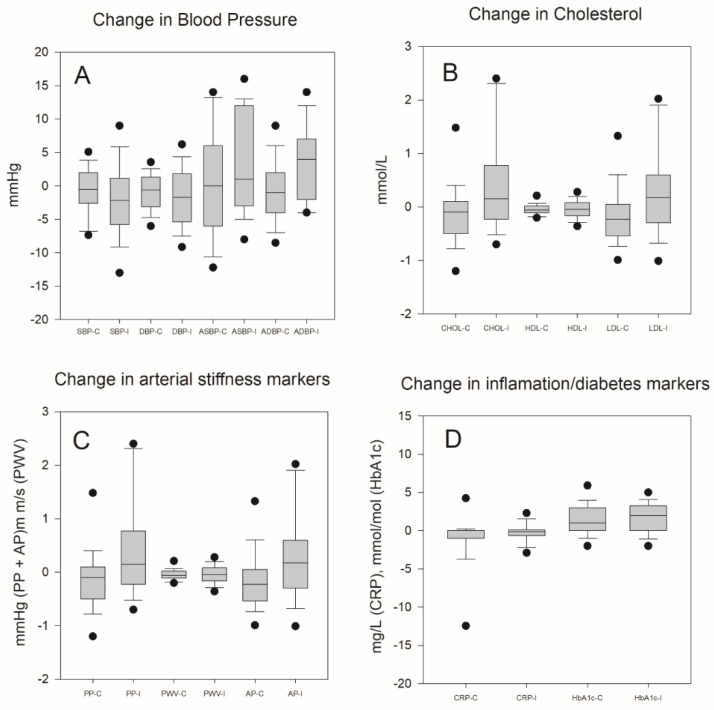
Changes in BP (**A**), CHO (**B**), markers of arterial stiffness (**C**), and (**D**) markers of inflammation and diabetes from baseline to post-intervention. Horizontal line inside box is median, whiskers are 5 and 95 percentiles, dots outliers. SBP = systolic blood pressure, DBP = diastolic blood pressure, ASBP = aorta systolic blood pressure, ADBP = aorta diastolic blood pressure, PP = pulse pressure, AP = augmented pressure, CRP = C-reactive protein, CHOL = total cholesterol, HDL = high-density lipoprotein, LDL = low-density lipoprotein, HbA1c = glycated hemoglobin, C = control group, I = intervention group. Dots are outliers.

**Table 1 ijerph-17-03943-t001:** Demographic characteristics among shift workers (N = 65) participating in the study.

Variables	Shift Workers Plant A (N = 42)			Shift Workers Plant B (N = 23)		
	Number	Mean	SD	Number	Mean	SD
Age (years)		40.5	11.0		40.0	12.5
Women	5			1		
BMI (kg/m^2^)		27.2	5.2		26.4	4.4
Pack-years		9.2	14.4		5.0	8.0
Daily smokers	11			3		
College/University	2			1		
No. of years as shift worker		14.5	10.1		15.0	9.9
Physical activity, high intensity (min)		90.1	145		67.0	69.0

**Table 2 ijerph-17-03943-t002:** Baseline data for the control and intervention groups.

Group	Variable	Minimum	Maximum	Mean	SD
Control	Age (yr)	21.0	57.0	38.4	11.5
	Height (cm)	160.0	198.0	181.8	7.3
	Body Mass (kg)	55.0	130.0	89.8	19.1
Intervention	Age (yr)	28.0	58.0	43.1	10.5
	Height (cm)	155.0	190.0	175.5	9.2
	Body Mass (kg)	48.0	123.0	82.8	18.8

**Table 3 ijerph-17-03943-t003:** Baseline and post-intervention results for control and intervention groups.

Variables	Control				Intervention			
	n	Mean	SD	SEM	n	Mean	SD	SEM
BL BM (kg)	31	88.8	18.6	3.3	19	82.8	18.8	4.3
PO BM (kg)#	31	89.5 *	18.7	3.4	19	81.7	17.8	4.1
BL SBP (mmHg)	30	126.0	14.5	2.6	19	124.4	10.6	2.4
PO SBP (mmHg)	31	124.2	14.3	2.6	19	120.1	12.3	2.8
BL DBP (mmHg)	30	82.4	7.6	1.4	19	83.6	6.4	1.6
PO DBP (mmHg)	31	80.6	9.3	1.7	19	80.3	9.5	2.2
BL ASBP (mmHg)	31	111.3	13.0	2.3	19	112.8	10.2	2.4
PO ASBP (mmHg)	31	109.2	18.3	3.3	19	109.3 *	8.4	1.9
BL ADBP (mmHg)	31	72.0	11.8	2.1	19	75.2	8.1	1.9
PO ADBP (mmHg)	30	73.8	8.8	1.6	19	72.2 *	8.0	1.8
BL PP (mmHg)	31	38.1	6.2	1.1	19	37.6	5.6	1.3
PO PP (mmHg)	31	37.9	8.5	1.5	19	37.1	4.5	1.0
BL PWV (m/s)	29	7.6	1.4	0.3	18	7.7	1.4	0.3
PO PWV (m/s)	30	7.8	1.8	0.3	19	7.9	1.3	0.3
BL AP (mmHg)	31	9.0	5.7	1.0	19	10.0	3.9	0.9
PO AP (mmHg)	31	8.1	4.8	0.9	19	9.4	3.6	0.8
BL CRP (mg/L)	31	1.6	2.4	0.4	19	2.3	2.6	0.6
PO CRP (mg/L)	30	2.7	4.5	0.8	18	2.1	1.9	0.4
BL CHOL (mmol/L)	31	4.7	0.8	0.1	19	5.2	0.9	0.2
PO CHOL (mmol/L)#	30	4.9	1.0	0.2	18	4.8	0.8	0.2
BL HDL (mmol/L)	31	1.2	0.2	0.0	19	1.3	0.4	0.1
PO HDL (mmol/L)	30	1.2	0.2	0.0	18	1.3	0.4	0.1
BL LDL (mmol/L)	31	3.0	0.7	0.1	19	3.3	0.9	0.2
PO LDL (mmol/L)	30	3.2	0.9	0.7	18	3.1	0.9	0.2
BL HbA1c (mmol/mol)	31	34.3	3.9	0.7	19	34.4	4.0	0.9
PO HbA1c (mmol/mol)	30	32.8 **	4.6	0.8	18	32.5 **	3.5	0.8
BL VPA (min/week)	31	69.0	73.5	13.2	19	85.8	126.8	29.0
PO VPA (min/week)	29	84.2	118.9	22.1	18	106.2	53.0	12.5
BL r$\dot{V}$O_2max_ (mL/kg/min)	11	44	9	3	14	44	8	2
PO r$\dot{V}$O_2max_ (mL/kg/min)	11	45	9	3	14	44	7	2
BL a$\dot{V}$O_2max_ (mL/min)	11	3609	523	158	14	3509	436	117
PO a$\dot{V}$O_2max_ (mL/min)	11	3646	381	115	14	3489	469	125

BM = body mass, SBP = systolic blood pressure, DBP = diastolic blood pressure, ASBP = aorta systolic blood pressure, ADBP = aorta diastolic blood pressure, PP = pulse pressure, AP = augmented pressure, CRP = C-reactive protein, CHOL = total cholesterol, HDL = high-density lipoprotein, LDL = low-density lipoprotein, HbA1c = glycated hemoglobin, VPA = vigorous physical activity, r$\dot{V}$O_2max_ = maximal oxygen uptake relative to body mass, a$\dot{V}$O_2max_ = maximal oxygen uptake. BL = baseline, PO = post-intervention. * = *p* < 0.05, ** = *p* < 0.01 between BL and PO values, # = *p* < 0.05 between C and I changes.

**Table 4 ijerph-17-03943-t004:** Case characteristics and selected outcomes in a young man with more than 10 years as a shift worker performing 2–3 recorded weekly PA interventions during follow-up.

	Baseline	Post-Intervention
BMI (kg/m^2^)	34.7	34.0
RHR ^a^ (bpm)	73	55
SBP ^a^ (mmHg)	133	114 ^b^
DBP ^a^ (mmHg)	90	72 ^b^
ASBP (mmHg)	121	109 ^b^
ADBP (mmHg)	76	69 ^b^
AP (mmHg)	11	8
AIx	25	19 ^b^
PWV (m/s)	9.0	7.9
CRP (mg/L)	6.2	7.5
Cholesterol/HDL (mmol/L)	4.2	4.0
HbA1c (mmol/mol)	32	30
$\dot{V}$O_2_max (ml/kg/min)	33.8	34.9

^a^ Best of three measurements after 5 min rest; ^b^ clear health improvement, below cut-off values; BMI = body mass index, RHR = resting heart rate, Aix = augmentation index (see Abbreviations).

**Table 5 ijerph-17-03943-t005:** Number of people that have moved from having an unhealthy value to a healthy value during the intervention.

Variable	Control	Intervention
CHOL	−4	2
HDL	2	0
LDL	1	2
CRP	−3	0

CHOL = total cholesterol, HDL = high-density lipoprotein, LDL = low-density lipoprotein, CRP = C-reactive protein. A – sign indicates a decrease in number of people in the healthy group. HbA1c not shown because all participants were in the healthy group at both times.

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
