# Peer review of "High-Intensity Training Reduces CVD Risk Factors among Rotating Shift Workers: An Eight-Week Intervention in Industry"

_ijerph, 2020, doi:10.3390/ijerph17113943_

Round 1

Reviewer 1 Report

This paper introduces an interesting study showing the effects of physical activity and intense training on the health of rotating shift workers.

The topic and the results are interesting, the paper is well-structured and the methodology is rigorous.  

Few suggestions for the authors:

Introduction.

Add a paragraph describing the research questions that you investigated during the study.

Check abbreviations (e.g. P2L49 PA is not defined in the introduction)

Material and method

Add a table describing the characteristics of intervention and control groups. It could be placed after the selection process in Figure 1. 

Conclusions

In its current version, this paragraph does not add any value to the paper. The authors should use this section to remind the main results of the study, describing the contribute of this study to the literature and the research on this topic. Finally, you may provide the answers to the research questions introduced in the first section. 

Add an Appendix with the all the abbreviations adopted in the paper.

Author Response

We thank You for the comments! We have no objections, and have tried to include them in the revised manuscript. 

Reviewer 1

This paper introduces an interesting study showing the effects of physical activity and intense training on the health of rotating shift workers.

The topic and the results are interesting, the paper is well-structured and the methodology is rigorous.  

Few suggestions for the authors:

Introduction.

Add a paragraph describing the research questions that you investigated during the study.

Aim of the study: Given our results of CVD-risk factors among shift-workers [12] and a worry that shift-workers tend to exercise less than other workers [6], the Occupational Health Service wanted to study if high intensity PA could modify the risk of early manifestations of CVD in this group of industrial workers

Check abbreviations (e.g. P2L49 PA is not defined in the introduction)

Deleted

Material and method

Add a table describing the characteristics of intervention and control groups. It could be placed after the selection process in Figure 1. 

Table 2: Baseline data for the Control and Intervention groups

Group

Variable

Minimum

Maximum

Mean

SD

Control

Age (yr)

21.0

57.0

38.4

11.5

Height (cm)

160.0

198.0

181.8

7.3

Body Mass (kg)

55.0

130.0

89.8

19.1

Intervention

Age (yr)

28.0

58.0

43.1

10.5

Height (cm)

155.0

190.0

175.5

9.2

Body Mass (kg)

48.0

123.0

82.8

18.8

Conclusions

In its current version, this paragraph does not add any value to the paper. The authors should use this section to remind the main results of the study, describing the contribute of this study to the literature and the research on this topic. Finally, you may provide the answers to the research questions introduced in the first section. 

Conclusions

A PA-intervention focusing on cardiorespiratory health among rotating-shift-workers in industry, indicates that PA modifies early manifestations of CVD. This could imply that regular, short bouts of PA could counteract the increased risk for coronary heart disease and stroke among these workers. PA-interventions in occupational settings may thus decrease coronary heart disease and stroke incidences in this vulnerable group of workers. The present cohort will be subjected to follow-up for three years.

Add an Appendix with the all the abbreviations adopted in the paper.

Added

Reviewer 2 Report

The present study by Mamen et al. has studied the effect of 17 min high-intensity training three times a week among shift-workers for about two months on CVD risk factors. The study was done on sixty-five shift workers and at the initial phase, they were all healthy. Finally, data were observed on 56 workers. Out of 56 worker, 37 designated as control group and 19 participants were in the intervention group.

A questionnaire was prepared for the workers related to the Physical Activity (PA) before and after the training intervention. The study measured blood pressure, heart rate, lipids, glycated haemoglobin (HbA1c), and C-reactive protein (CRP) and arterial stiffness. After training, the reported study result was beneficial for the shift-workers and they reduced the CVD risk factors. Hence the study concluded by saying that PA-intervention can be useful in treating vulnerable groups of workers. However, there are few comments to be responded before finalizing the decision on the MS.

(1) Line 48, please define VO2 max
(2) Line 51-52, Please provide some reason for this.. Possibly, from other published articles.
(3) In “Occupational Medicine” O and M should be small letter
(4) Figure 1 the ray diagram is not impressive. Please draw it again with paramount information in it.
(5) Line 121-122, the abbreviation was wrong for systolic and diastolic blood pressure.
(6) Line 123, the line should be elaborate for what purpose
(7) Line 124, it is better to define the central blood pressure “you can define CBP, as this is new for many researchers”
(8) Line 129, Glycated haemoglobin (G should be small) and Line 132 (W should be small)
(9) Line 200, it is better to write ASBP instead of ABPS, in that way you need to modify into the table as well
(10) Figure 2 figure is blur. Please change this with the original one. Text Box should be redrawn.
(11) Figure 5, What the dot is expressing, the author should have to define in the figure legend.
(12) Line 268-269, Please explain the meaning of the sentence
(13) Line 307-309, Did the study check and verify the result of the inflammatory marker, I think it is needed to include in the discussion comprehensively.
(14) The possible reason should be present in the discussion, how the author discusses their present finding. Need to rewrite the discussion part.
(15) Line 316 to 322, repeated sentence.

Author Response

We appreciate Your comments! There are no objections from our side.

Reviewer 2

The present study by Mamen et al. has studied the effect of 17 min high-intensity training three times a week among shift-workers for about two months on CVD risk factors. The study was done on sixty-five shift workers and at the initial phase, they were all healthy. Finally, data were observed on 56 workers. Out of 56 worker, 37 designated as control group and 19 participants were in the intervention group.

A questionnaire was prepared for the workers related to the Physical Activity (PA) before and after the training intervention. The study measured blood pressure, heart rate, lipids, glycated haemoglobin (HbA1c), and C-reactive protein (CRP) and arterial stiffness. After training, the reported study result was beneficial for the shift-workers and they reduced the CVD risk factors. Hence the study concluded by saying that PA-intervention can be useful in treating vulnerable groups of workers. However, there are few comments to be responded before finalizing the decision on the MS.

  • Line 48, please define VO2 max

In addition, we assessed maximal aerobic power, V̇O2max [12].

(2) Line 51-52, Please provide some reason for this.. Possibly, from other published articles.

In this cross-sectional study of workers in industry, the number of years of shift work was associated with early manifestations of CVD, with an increased IMT in the carotid artery and increasing CRP. This could imply an increased risk for coronary heart disease and stroke among these workers. We will follow the present cohort for three years.

Other useful citations include, but since we have 41 references already, we are reluctant to add more, but can do so if You insist.

  1. Hublin C, Partinen M, Koskenvuo K, Silventoinen K, Koskenvuo M, Kaprio J. Shift-work and cardiovascular disease: a population-based 22-year follow-up study. Eur J Epidemiol. 2010 May;25(5):315–23.
  2. Puttonen S, Kivimäki M, Elovainio M, Pulkki-Råback L, Hintsanen M, Vahtera J, Telama R, Juonala M, Viikari JSA, Raitakari OT, Keltikangas-Järvinen L. Shift work in young adults and carotid artery intima-media thickness: The Cardiovascular Risk in Young Finns study. Atherosclerosis. 2009 Aug;205(2):608–13.
  3. Puttonen S, Härmä M, Hublin C. Shift work and cardiovascular disease - pathways from circadian stress to morbidity. Scand J Work Environ Health. 2010 Mar;36(2):96–108.
  4. Torquati L, Mielke GI, Brown WJ, Kolbe-Alexander T. Shift work and the risk of cardiovascular disease. A systematic review and meta-analysis including dose-response relationship. Scand J Work Environ Health. 2018 01;44(3):229–38.
  5. van LeeuwenWMA, Lehto M, Karisola P, Lindholm H, Luukkonen R, Sallinen M, Härmä M, Porkka-Heiskanen T, Alenius H. Sleep restriction increases the risk of developing cardiovascular diseases by augmenting proinflammatory responses through IL-17 and CRP. PloS One. 2009;4(2):e4589.

  • In “Occupational Medicine” O and M should be small letter

Actualy that is the Name of the journal, but the line has been adjusted.

(4) Figure 1 the ray diagram is not impressive. Please draw it again with paramount information in it.

All eligible shift-workers: N=106

Plant A: n=51

Plant B: n=55

Baseline registrations: N=65

Plant A: n=42

Plant B: n=23

8-week follow-up: N=56

Plant A: n=36

Intervention group with ≥10 training sessions: 19

Control group with < 10 training sessions: 17

Plant B: n=20

Intervention group: n=19

Control group: n=17+20=37

106 shift-workers were available for the project, 51 from Plan A and 55 from Plan B. At project start 42 shift-workers from Plant A accepted the invitation with three weekly trainings sessions, with 23 workers from Plant B as “activity as usual” controls. At follow-up eight weeks later, 19 participants from the intervention group had participated in at least 10 training sessions and were included in the analyses as intervention group. The remaining 17, who had less than 10 training sessions logged, were put in the control group, together with the 20 workers from Plant B, giving 37 in this group.

(5) Line 121-122, the abbreviation was wrong for systolic and diastolic blood pressure.

Corrected

(6) Line 123, the line should be elaborate for what purpose

While the participant was sitting, we measured blood pressure and resting heart rate (RHR) to look for signs of cardiovascular disease. The measurements took place after five minutes of rest, on the left arm three times in intervals of one minute in which the mean value of three measurements on the systolic (BPS) and diastolic pressure (BPD) were used in the statistical analysis at both occasions. Using the mean is the current practice at our laboratory,

(7) Line 124, it is better to define the central blood pressure “you can define CBP, as this is new for many researchers”

We assessed central blood pressure (CBP) which is the pressure in ascending aorta, augmentation pressure (AP), augmentation index (AIx), pulse pressure (PP) and pulse wave velocity (PWV) with a SphygmoCor XCEL® (AtCor Medical Pty Ltd., Sydney, Australia).

(8) Line 129, Glycated haemoglobin (G should be small) and Line 132 (W should be small)

Corrected

(9) Line 200, it is better to write ASBP instead of ABPS, in that way you need to modify into the table as well

Changed to ASBP/ADBP, and also BPS to SBP and so on

(10) Figure 2 figure is blur. Please change this with the original one. Text Box should be redrawn.

Box 1. Case characteristics and selected outcomes in a young man with more than 10 years as a shift-worker performing 2-3 recorded weekly PA-interventions during follow-up

Baseline

Post intervention

BMI (kg/m2)

34.7

34.0

RHRa (bpm)

73

55

BPSa (mmHg)

133

114b

BPDa (mmHg)

90

72b

ASBP (mmHg)

121

109b

ADBP (mmHg)

76

69b

AP (mmHg)

11

8

AIx

25

19b

PWV (m/s)

9.0

7.9

CRP (mg/L)

6.2

7.5

Cholesterol/HDL (mmol/L)

4.2

4.0

HbA1c (mmol/mol)

32

30

V̇O2max (ml/kg/min)

33.8

34.9

aBest of three measurements after 5 min rest                               

bClear health improvement, below cut-off values

BMI=Body Mass Index, RHR=Resting Heart Rate, AIx= augmentation index, else see Table 2 for abbreviations.

(11) Figure 5, What the dot is expressing, the author should have to define in the figure legend.

Dots are outliers added to the text. Now Figure 4, figure 3 deleted

(12) Line 268-269, Please explain the meaning of the sentence
Changed to: As the criterium for being admitted to the Intervention Group was only to have attended at least 10 sessions, the training effect in that group may have been low.

(13) Line 307-309, Did the study check and verify the result of the inflammatory marker, I think it is needed to include in the discussion comprehensively.

Yes, we used CPR as the only inflammation marker, it did not change much during the project. Probably due to the short intervention duration. .

(14) The possible reason should be present in the discussion, how the author discusses their present finding. Need to rewrite the discussion part.

We have tried to re-write, see below

(15) Line 316 to 322, repeated sentence.

      Systemic inflammation expressed as elevated high sensitivity-CRP is known to increase vascular risk [38]. Shift work could cause sleep deprivation which again affects the immune system resulting in increased inflammation [6,7]. This phenomenon could explain that number of years as a shift-worker is associated with increasing CRP-levels initially in the present cohort [8]. Leisure time PA, among workers with long working hours , however, resulted in reduced CRP after 15-months of follow-up [8].  The present 8-week PA-intervention did not affect this parameter.  Thus, the duration of this PA-initiative might be too short to reduce inflammation among shift workers. 

Round 2

Reviewer 2 Report

No further comments. 

Author Response

We thank You again for Your comments, and we have included all of them in this new revision.
